# The Intake of Phosphorus and Nitrites through Meat Products: A Health Risk Assessment of Children Aged 1 to 9 Years Old in Serbia

**DOI:** 10.3390/nu14020242

**Published:** 2022-01-06

**Authors:** Jelena Milešević, Danijela Vranić, Mirjana Gurinović, Vladimir Korićanac, Branka Borović, Milica Zeković, Ivana Šarac, Dragan R. Milićević, Maria Glibetić

**Affiliations:** 1Centre of Research Excellence in Nutrition and Metabolism, Institute for Medical Research, National Institute of Republic of Serbia, University of Belgrade, Tadeusa Koscuska, 111000 Belgrade, Serbia; jelena.milesevic@gmail.com (J.M.); mirjana.gurinovic@gmail.com (M.G.); zekovicmilica@gmail.com (M.Z.); ivanasarac@yahoo.com (I.Š.); mglibetic@gmail.com (M.G.); 2Institute of Meat Hygiene and Technology, Kaćanskog 13, 11040 Belgrade, Serbia; danijela.vranic@inmes.rs (D.V.); vladimir.koricanac@inmes.rs (V.K.); branka.borovic@inmes.rs (B.B.)

**Keywords:** nitrite, phosphorus, food additives, dietary intake, children, Serbia

## Abstract

This study provides the data on dietary exposure of Serbian children to nitrites and phosphorus from meat products by combining individual consumption data with available analytical data of meat products. A total of 2603 and 1900 commercially available meat products were categorized into seven groups and analysed for nitrite and phosphorous content. The highest mean levels of nitrite content, expressed as NaNO_2_, were found in finely minced cooked sausages (40.25 ± 20.37 mg/kg), followed by canned meat (34.95 ± 22.12 mg/kg) and coarsely minced cooked sausages (32.85 ± 23.25 mg/kg). The EDI (estimated daily intake) of nitrites from meat products, calculated from a National Food Consumption Survey in 576 children aged 1–9 years, indicated that the Serbian children population exceeded the nitrite ADI (acceptable daily intake) proposed by EFSA (European Food Safety Authority) in 6.4% of children, with a higher proportion in 1–3-year-old participants. The mean phosphorus concentration varied from 2.71 ± 1.05 g/kg to 6.12 ± 1.33 g/kg in liver sausage and pate and smoked meat products, respectively. The EDI of phosphorus from meat products was far below the ADI proposed by EFSA, indicating that the use of phosphorus additives in Serbian meat products is generally in line with legislation.

## 1. Introduction

Results of the latest Global Burden of Diseases, Injuries, and Risk Factors Study (GBD) from 2019 have shown that the diet is still one of the most important risk factors for attributable mortality and disability-adjusted life years (DALYs) [1]. Meat and processed meat products (ham, sausages, bacon, frankfurters, salami, etc.) are often perceived as unhealthy by consumers due to their intakes having been positively associated with the risk of several major chronic diseases. According to several reports of the World Cancer Research Fund (WCRF) [2], there is convincing evidence that the consumption of red and particularly processed meat is associated with cancer risk. Additionally, the International Agency on Research on Cancer (IARC) has classified processed meat as carcinogenic to humans (Group 1 carcinogen), based on sufficient evidence in humans that the consumption of processed meat causes colorectal cancer (CRC) and that red meat consumption is probably carcinogenic to humans (Group 2A) [3]. Red meat and processed meats contain multiple substances that are potentially carcinogenic, including nitrates, nitrites, polycyclic aromatic hydrocarbons (PAHs), and heterocyclic amines (HCAs), resulting from cooking or processing. 

Food additives are intentionally added to food products for a technological purpose in the manufacture, processing, preparation, treatment, packaging, transport, or storage of food. Nitrates and nitrites, in the form of sodium and potassium salts, are widely used as preservatives in meat production (E249–E252). From the technological point of view, the main reason for adding nitrites and/or nitrates in the processing of meat products is to improve the quality (stabilize red meat colour and texture; may also contribute to the product flavour characterization) [4,5,6] and durability, due to retardation of the oxidative rancidity [7]; the additions and may also contribute to the safety of products by inhibiting the growth and reproduction of bacteria *Clostridium botulinum, Clostridium perfringens, Staphylococcus aureus*, and *Bacillus cereus* [7,8]. They are also used in other processed foods, such as cheese and fish, for preservation purposes [9]. Besides this, nitrites and nitrates are naturally present in vegetables [10], and they are in water as residues of contamination of ground water and surface water as a result of manuring and fertilisation practices.

Although nitrites and nitrates are not carcinogenic, due to possible formation of carcinogenic nitrosamines under certain conditions, a diet high in nitrates and nitrites is associated with increased risk of colorectal cancers (CRC) [2,3,11]. Ingested nitrate is reduced to nitrite by the bacterial flora in the mouth and digestive tract. Further, nitrite reacts with amines, amides, or amino acids and other nitrosation precursors in the gastrointestinal tract to form carcinogenic *N*-nitroso compounds (NOCs) [12,13]. Carcinogenic *N*-nitroso compounds can be found in many processed types of meat and can be endogenously formed after ingesting red meat in the human intestines by the bacterial flora. Because endogenous nitrosation is estimated to account for 45–75% of total NOC exposure [14], dietary intake of nitrate and nitrite, precursors for endogenous nitrosation, may be important colorectal cancer risk factors. There is strong evidence of an association between red and processed meat consumption and the risk of colorectal cancer [15,16]. In addition, processed meat products, and especially cured meat, already contain preformed NOCs [17]. Therefore, they are the major source of nitrites and N-nitrosamines in human dietary intake. Another aspect of CRC is the fact that red meat is also the source of heme iron, which participates in the abovementioned process of endogenous N-nitrosation in the intestine. Moreover, that nitrite could oxidize haemoglobin to methaemoglobin, which cannot bind and transport oxygen to tissues, leading to acute or chronic toxicity such as methemoglobinemia [18,19]. Nitrate also competitively inhibits iodide uptake by the thyroid [20,21], possibly affecting thyroid hormone production and potentially resulting in thyroid tumour promotion. Thus, the World Cancer Research Fund and the American Institute for Cancer Research (WCRF/AICR) recommend consuming <500 g/week of red meat and <50 g/day of processed meat [16].

The current Serbian legislation has restricted the maximum amount of nitrate or nitrite that may be added in processed meat expressed as NaNO_2_ or NaNO_3_ to 100 and 150 mg/kg, depending on the type of product [22,23], whereas regulations in Europe state that the maximum amount of nitrite and nitrates that may be added to the processed meat during manufacturing should be from 50 to 180 mg/kg, with a number of exemptions [19]. The Joint Expert Committee on Food Additives (JECFA) established an acceptable daily intake (ADI) for nitrite of 0–0.07 mg per kg body weight per day (mg/kg bw/day), expressed as nitrite ion based on a no-effect level (NOEL) of 6.7 mg/kg bw/day for effects on the heart and lungs in a 2-year study in rats, and a safety factor of 100 [19,24,25].

Among the food additives, food-grade phosphates are widely used as an additive compound in various products. Phosphorus additives (E338–341, E343, E450–452) are increasingly being used in processed and fast foods, especially in the meat industry, cheeses, baked products, and beverages for several technological purposes. They increase water holding capacity (WHC), preserve moisture or colour, emulsify ingredients, and enhance flavour, as well as stabilize foods. Despite its technological benefits, it has been estimated that 50% of daily phosphorus (P) intake in the Western world is from “hidden phosphorus” as a food additive [26]. Furthermore, phosphorus in food additives is rapidly and almost completely absorbed, whereas a natural constituent of protein-bound phosphorus is more slowly and less efficiently absorbed (60%) [27]. An association between excessive phosphate intake, high serum phosphate levels (hyperphosphatemia), and cardiovascular morbidity and mortality in patients with chronic kidney disease and bone health complications has long been known [28]. Due to the increased consumption of processed foods, high phosphorus intake from additives should be taken into account as a potential public health concern. Thus, phosphates are included in the list of food additives that must be reduced in meat preparations. These additives were recently evaluated by the Scientific Committee for Food [28], which derived a group acceptable daily intake (ADI) for phosphates expressed as phosphorus of 40 mg/kg bw/day. The Panel concluded that this ADI is safe for healthy adults because it is below the doses at which clinically relevant adverse effects were reported in short-term and long-term studies in humans. In addition, European Directives on food additives [29] require that Member States monitor intakes to ensure that consumers do not have an excessive intake of a given food additive, which could lead to health hazards. The Serbian standard maximum limit for total phosphates expressed as P_2_O_5_ in meat products is <8 g/kg [22] or ≤5 g/kg of added phosphorus [23].

Chronic noncommunicable diseases such as cardiovascular diseases and cancer are a national public health problem and are the leading causes of death in Serbia (47.3% and 17.8%, respectively). In Serbia, they constitute the major contributor to the burden of disease in terms of DALYs or mortality [30]. Meat products such as ham, sausages, bacon, frankfurters, salami, etc., are widely consumed by all groups of the population in Serbia, at home or in fast foods restaurant. Although comprehensive national surveys of assessment of total dietary phosphorus intake and food additives such as nitrates and nitrites in Serbia are scarce, studying the nutritional status as well as a lifestyle of the population, particularly children, is fundamental to design national guidelines and public health policies. 

Therefore, the objective of this study was to determine nitrites and phosphorus content in processed meat products. Based on the analysis results, dietary exposure of the Serbian children population to nitrites and phosphorus was then estimated and discussed.

## 2. Materials and Methods

### 2.1. Reagents 

All chemicals and reagents used were of analytical reagent grade from Merck Co. (Darmstadt, Germany) unless otherwise stated.

### 2.2. Meat Products and Sample Preparation

In the present study, a total of 2603 meat product samples, produced by the Serbian meat industry or imported (241 bacon, 362 canned meat, 353 coarsely minced cooked sausages, 683 dry fermented sausages, 580 finely minced cooked sausages, 46 liver sausage and pate, and 338 smoked meat products), were obtained from different regions from the Serbian retail market during 2018–2021 and were analysed for nitrite content. In most of the meat products, all parameters of quality defined by the legislation were examined, and in a smaller number, analyses were carried out as per the client’s request.

In the same period of investigation, a total of 1900 meat product samples, categorized into six groups including bacon, canned meat, coarsely minced cooked sausages, finely minced cooked sausages, liver sausage and pate, and smoked meat products were analysed for phosphorous content. 

All meat products were kept at refrigeration temperature and analysed within 48 h. If the analyses were not conducted on the same day, the samples were stored in a refrigerator at 4 °C until required for testing.

The analysed samples were thawed and blended in a commercial kitchen blender unit (Homogenizator Blixer 2, Robot Coupe, Vincennes, France (2.9 L) 700 w, 3000 rpm). For each sample, two composite samples were prepared. All samples were then analysed in duplicate.

### 2.3. Determination of Nitrite Content

The content of nitrite in examined meat products was determined according to the standard ISO procedure [31]. A representative sample amount (~10 g) was then measured into 300 mL flask using an analytical balance (Mettler, AE 200, USA), followed by the addition of a solution of hydrous sodium borate, Na_2_B_4_O_7_.10H_2_O (50 g/L) and 100 mL deionised water at 70.0 ± 0.2 °C. Residual nitrite extraction was achieved by keeping the samples in a hot water bath, at the temperature of boiling, for 15 min, and every 5 min, flasks were shaken vigorously. After cooling, 2 mL of each Carrez solution (Carezz reagent I and Carezz reagent II) were added and mixed thoroughly. Samples were then diluted to 200 mL with deionized water. Samples were filtered through quantitative cellulose filters (pore size < 5 µm). Colour generation was achieved by transferring an aliquot of the filtrate (25 mL) to a 100 mL volumetric flask and adding 10 mL of the sulphanilamide solution and then 6 mL conc. HCl. Flasks were stored in the dark for 5 min. Subsequently, 2 mL solution of N-naftil-1-ethylenediamine-chloride (0.25 g/250 mL) was added to each flask and moved to the dark for 3 min. Thereafter, samples were diluted to 100 mL. Absorbance was measured at 538 nm using a spectrophotometer (UV/VIS Spectrophotometer, Jenway 6405). A procedural blank was run with every batch of samples.

Calibration curves were generated using the concentration levels ranging from 2.5 to 10 NaNO_3_ µg mL^−1^, Y = 0.0669X + 0.024: *R*^2^ = 0.999. A recovery study of the analytical procedure was carried out by spiking several already analysed samples with standard solutions, and recovery rates were found to be between 87% and 94%. The nitrite content is expressed as NaNO_2_ (mg·kg^−1^), following: c × 2000/m × V, where c is the concentration of NaNO_2_ (µg/mL) from the calibration curve, m is the mass of sample (g) for analysing, and V is a volume of an aliquot of the filtrate used for spectrometric determination.

### 2.4. Phosphorus Measurement

The total phosphorus content, expressed as P_2_O_5_ (g/kg), in examined meat products was determined according to the standard ISO procedure [32]. In brief, ~5 g portion of samples (measured using an analytical balance (Mettler, AE 200, USA) was ashed at the maximum temperature of 500 °C in a muffle furnace (LE 14/11/R7, Nabertherm). On completion of the digestion, the white ash was dissolved by heating with dilute nitric acid (1 + 1, *v*/*v*) and quantitatively transferred in 100 mL flask. Then, made up by the addition of deionized water, and after mixing, the solution was then filtered, and the first 5 to 10 mL were discarded.

Aliquots (20 mL) of treated solution were pipetted into 100 mL volumetric flasks and mixed thoroughly with 30 mL ammonium heptamolybdate solution 50g L^−1^. The resulting solution was then diluted to the volume with deionized water. After 15 min at room temperature, the absorbance was read against a reagent blank at 430 ± 2 nm using a UV-visible spectrophotometer (UV/VIS Spectrophotometer, Jenway 6405). The standard curve was determined under the same conditions as those for the samples using potassium dihydrogen phosphate as a standard (10–60 P_2_O_5_ µg/mL; Y = 0.0187X−0.0096: *R*^2^ = 0.9999). A recovery study of the analytical procedure was carried out by spiking several already analysed samples with standard solutions, and recovery rates were found to be between 89% and 95%. The total phosphorus content, expressed as P_2_O_5_ (g/kg), following: c/20 m, where c is the concentration of P_2_O_5_ (µg/mL) from the calibration curve, and m is the mass of the sample (g) for analysis.

### 2.5. National Food Consumption Survey on Toddlers and Children

The National Food Consumption Survey on toddlers and children, in compliance with the EFSA EU Menu methodology [33,34], was conducted between 2017 and 2021 and included a total of 576 participants, comprising 290 toddlers aged from 1 to 2.9 years old and 286 children aged from 3 to 9 years old. Data collected included: general questionnaire, body weight and height measurements, age-appropriate Food Propensity Questionnaire (FPQ), and twice repeated 24 h food record. The consumed portion sizes were estimated based on natural units, household measures, packaging information, and a validated national Food Atlas for Portion Size Estimation [35]. Frequency of consumption of processed meat products was explored by FPQ, which categorized consumption in seven frequency groups (never, less than once a month, 1–3/month, once per week, 2–3/week, 4–5/week, 6–7/week). Data were analysed using a nutritional software tool DIET ASSESS & PLAN (DAP) [36] and the Serbian Food Composition Database, developed in compliance with EuroFIR standards [37]. Weight measurements were performed without shoes and jackets using a digital balance where data were recorded to the nearest 0.1 kg or from the most recent paediatric report not older than 3 months. Minimum sample size (*n* = 130) was determined by EFSA EU Menu methodology [38]. The study sample distribution according to age and gender and body weight expressed as a mean value with standard deviations is shown in Table 1.

### 2.6. Exposure Assessment and Risk Characterization

Exposure was calculated for all the processed meat categories and both study groups according to their gender and age by the deterministic approach involving the average probable daily intake (APDI) method [39]. Nonbrand loyal scenario as the most relevant exposure scenario for the safety evaluation of phosphates was used [28]. NaNO_2_ concentrations were first converted to nitrite (66.65% of NaNO_2_), then multiplied by the processed meat consumption rate (kg/day) and divided by the average body weight. Results are shown in Section 3.

The same pattern was used for phosphorus content where P content was 43.64% of P_2_O_5_. Results are also shown in Section 3. Within the general framework of chemical risk assessment, a difficult step in dietary exposure evaluation is handling concentration data reported to be below the limit of detection (LOD). These data are known as nondetects, and the resulting occurrence distribution is left-censored. The left-censored data (data below LOD and LOQ) were processed by applying the substitution method of EFSA [40]. According to this guidance, for dietary exposure assessments, two exposure scenarios were considered. 

The mean values and 95th (P95) percentiles of phosphorus and nitrite content of the total dataset and processed meat categories were used to estimate daily intakes. The intake was calculated as the average daily intake for each studied group and expressed in mg/kg bw/day. The 95th (P95) percentiles of phosphorus and nitrite content were used to calculate the highest exposure scenario [41].

To evaluate the adequacy of intakes, the calculated estimated daily intake (EDI) of nitrites and total phosphorus were compared with the ADIs proposed by the EFSA [19,28].

### 2.7. Statistical Analysis

For statistical evaluation on data, Minitab 17 Ink statistical software was used (Minitab Ink., Coventry, UK). The results are presented as mean and standard deviation of the mean (SD).

## 3. Results and Discussion

### 3.1. Processed Meat Products Consumption among Children 1–9 Years Old

In Table 2 is given the average daily consumption of specific meat products according to 24 h food record data from the national survey. 

The average cumulative daily consumption of processed meat products per child was 20.31 ± 58.35 g/day, but it varied between the age groups: in younger children it was 14.39 ± 46.31 g/day, while in the older group it was 26.32 ± 61.97 g/day, almost doubled, with negligible difference between males and females (Table 2).

According to data from 24 h food records, 386 (i.e., 67.01%) participants consumed meat products in this survey. The percentage of consumers was higher in the older children group (3–9 years): 76.87% for males and 79.86% for females, compared with the younger children group: 54.36% for males and 57.45% for females.

Additional analysis determined frequency of consumption of meat products within the children’s groups. For this analysis, the FPQ records were used, which documented a frequent consumption of processed meat products (i.e., 2–3/week and more) in 62.67% of children, in total (Figure 1). However, the frequency was higher in the older group, where 73.47% and 73.38% of males and females, respectively, were frequent consumers, while in the younger group, 51.01% and 53.19% of males and females, respectively, were frequent consumers. These data are in accordance with the results obtained from 24 h food records.

In all child groups, the highest consumption was noted for finely minced cooked sausages and canned meat, followed by bacon and liver sausages and pate (Table 2).

### 3.2. Nitrite Content in Processed Meat Products

The nitrite content for the 2603 meat products used in the exposure assessment is presented in Table 3, while the frequency distribution categorized by product type is given in Figure 2. The overall nitrite detection rate was 85.97%, with nitrite detected in 2238 samples of the seven categories of meat products. Among the seven food categories, finely minced cooked sausages had the highest detection rate (99.6%) and the highest overall mean concentration of nitrite at 40.25 ± 20.37 mg/kg, followed by canned meat, coarsely minced cooked sausages, smoked meat products, bacon, liver sausage and pate, and dry fermented sausages, with overall mean concentrations of 34.95, 32.85, 23.09, 15.26, 12.62, and 1.50 mg/kg, respectively. In terms of nitrite content of these products, heterogeneous results were obtained (Figure 2 and Table 3). A high standard deviation, almost the same as the mean value (23.31 ± 23.75 mg/kg), was caused by a high nitrite content distribution, which ranged from 0.05–180.25 mg/kg. Smoked meat product samples had the highest residual nitrite content (180.25 mg/kg) within the studied meat samples, followed by coarsely minced cooked sausages (113.51 mg/kg), finely minced cooked sausages (101.19 mg/kg), bacon (100.38 mg/kg), and canned meat 93.45 (mg/kg). The lowest residual nitrite content was found in dry fermented sausages (24.88 mg/kg). Following the Serbian and EU regulations [29], the compliance levels were estimated at 99.9%. One sample exceeded maximum limits for the addition of nitrite (in the category of smoked meat products, 180 mg/kg). The results obtained in the present study are very similar to the information available in the literature [42,43], where it was reported a 91% detection rate for nitrite in processed meats. Concerning overall mean residual nitrite content (23.31 ± 23.75 mg/kg), this level is comparable to the median level of 27 mg/kg nitrite in meat products reported in a German food monitoring study [44].

The levels of residual nitrite and nitrate in processed meat products are variable depending on the time and temperature used during processing and storing, the initial addition of nitrite and nitrate, the composition of the meat, pH, addition of antioxidant components such as ascorbate, and the presence of microorganisms [4,45]. Honikel [4] estimated that the decline in nitrite levels due to heating during manufacturing is about 35% of the added level, and thereafter, there is a continuing decrease in nitrite levels during storage. The results obtained show distributions of nitrites differed in the four product types (Figure 2), which is related to the stability of the nitrite content in the different meat products, i.e., is the consequence of their differing preparation and processing methods [46]. Furthermore, different cured meat products may require a different ratio of nitrite and nitrate as preservatives. In fermented sausages, the content of nitrite decreased during the ripening of sausages as a result of the process that takes place in the sausage, i.e., reduction of nitrite content is significant where the main process is nitrite conversion into nitrates in the weak acid environment. In fermented sausages, the presence of nitrite becomes latent because the process is reversible and nitrates, under certain conditions, can revert into nitrites [47].

### 3.3. Estimated Dietary Intake of Nitrites from Processed Meat Products

Dietary intake of nitrites from processed meat product in the Serbian children population is estimated according to average daily intake of meat products in this population and the content of nitrites salts, i.e., nitrite ion (NO_2_), in these products. Daily intakes of nitrites were expressed as EDI in mg/kg bw/day and are presented in Table 4.

The total cumulative EDI for nitrite ion from all processed meat products for the whole population was 0.021 mg/kg bw/day, which contributed 29.52% to the ADI (Table 4). When the 95th percentile (P95th) of nitrite ion content in meat products was used to calculate EDI (representing the highest risk exposure scenario [41]), the total EDI for the whole population was 0.042 mg/kg bw/day, contributing to almost two-thirds (59.96%) of ADI. The total cumulative EDI for different child groups was similar, ranging from 0.019 mg/kg bw/day (in females 1–3 years old) to 0.023 mg/kg bw/day (in males 3–9 years old). In terms of contribution to the ADI, the contribution of nitrites from meat products varied from 27.46% of ADI (in females 1–3 years old) to 32.27% of ADI (in males 3–9 years old), and when the highest exposure scenario was considered (EDI P95th), it varied from 57.54% to 63.12% of ADI (Table 4). In general, there was no substantial difference in total EDI between different age groups, in spite of much lower consumption of processed meat products in the younger group, which can be explained by the lower body weight in the younger group and expression of EDI per kilogram body weight.

Table 5 presents the analysis of EDI in the consumer group (i.e., only consumers were included). As we already mentioned in Section 3.1, the percentage of consumers was higher in the older children group than in the younger children group (e.g., 76.87% vs. 55.90%). Moreover, the daily intake of processed meat products was almost doubled in the older children group (Table 2). Nevertheless, the total EDI for nitrite from processed meat products per consumer was higher (almost doubled) in the younger children group: 0.037 mg/kg bw/day for both males and females, compared with the older children group: 0.021 mg/kg bw/day for both males and females (Table 5). Similarly, contribution of nitrites from processed meat products to ADI was higher in the younger group: 53.13%, when mean EDI was considered (and 107.60%, when highest exposure scenario was considered), compared with the older group: about 30.48%, when mean EDI was considered (and 62.93%, when highest exposure scenario was considered). We did not observe significant differences between males and females in each age group (Table 4 and Table 5).

On average, in the whole child population (with both consumers and nonconsumers included), the percentage of those who exceeded ADI was 6.25% (and 17.01%, when the highest exposure scenario was considered). When only consumers were regarded, the percentage of those who exceeded ADI was 9.33% (i.e., 25.39%, when the highest exposure scenario was considered). Again, the proportion of children who exceeded ADI was higher in the younger group: 12.35% and 38.79%, for mean and highest exposure scenario, respectively, compared with 7.14% and 15.63%, in the older group. The reason for this discrepancy between the processed meat products intake and EDI per consumer in different age groups is that EDI is expressed per kilogram body weight (and body weight was much lower in the younger group) and that the proportion of consumers in younger group was lower. In all child groups, the highest contribution to total EDI derived from finely minced cooked sausages and canned meat (Table 4 and Figure 3), which is related to both the highest content of nitrites in those products (Table 3) and the highest consumption of those products by the children (Table 2).

We compared our results with the findings from other countries and surveys. Similar values of EDI in child populations were observed in many other studies. In the latest EFSA report [19] in which were given results from 15 EU countries (including Austria, Belgium, Bulgaria, Czech Republic, Denmark, Finland, France, Germany, Greece, Italy, Latvia, Netherlands, Spain, Sweden, and UK) involving 13,637 children, the mean EDI was 0.015 mg/kg bw/day in the 1–3 years old group, and 0.016 mg/kg bw/day in the 3–9 age group. The highest exposure for the 1–3-year-old group was noted in Denmark (0.027 mg/kg bw/day), and for the 3–9-year-old group was in Czech Republic (0.023 mg/kg bw/day). (For details on all included countries and surveys, please refer to the EFSA report) [19]. In a very recent Estonian study on children aged 1-10 years, mean nitrite intakes were 0.015 and 0.016 mg/kg bw/day and reached 21.9% and 22.9% of the ADI among children aged 1–3 years and 3–10 years, respectively. ADI was exceeded in 3.1% of children, predominantly in the younger age group [48]. In a study from Sudan [49], much higher intakes were noted (ranging from 0.026–0.128 mg/kg bw/day), despite much lower legal limits (100 mg/kg). In contrast, in another recent study in Korea, much lower nitrite intake was noted, which reached only 0.8% of the ADI for all subjects and 2.8% for the consumer group, which can be attributed to different dietary habits and significantly lower legal limits for nitrite content in processed meat products (70 mg/kg) [50]. 

In our study, 6.25% of all participants and 9.33% of consumers exceeded ADI. In the population of toddlers (1–3 years old), these proportions were even higher—12.35% (and reaching 40% at the highest exposure scenario). These values are alarmingly high, considering that nitrites in processed meat products are just one of numerous other natural and artificial sources of nitrites in the diet, including both food and water supplies. In the mentioned EFSA report [19], only 17% of total daily nitrite intake (i.e., EDI) is derived from food additives in processed meat products in the general population. Considering children, in the population of 1–3 years old, 8.9–27.0% of daily nitrite intake is derived from processed meat products, while in the population of 3–9 years old, 3.2–25.8% of daily nitrite intake is derived from these products [19]. The highest contribution of processed meat products for toddlers was in Denmark and, for other children, in Germany and Austria. A more recent Dutch study suggests that nitrates and nitrites in processed meat product account only for 8–9% of total daily intake of these compounds in the general population [10]. The majority of nitrates and nitrites in the diet derive from composite food, fruit and fruit products [19], vegetables and vegetable products, particularly leafy vegetables such as spinach and rucola [10], poultry, livestock meat, and cheese (and, in toddlers, additionally from food for infants and toddlers). Therefore, the EFSA Panel noted that the ADI would be exceeded for EU toddlers and children, both at the mean and at the highest exposure level, if all sources of dietary nitrite exposure (food additives, natural presence, and contamination) were considered [19]. Similarly, the actual EDI for nitrites in the Serbian children population might be much higher than proposed ADI when the whole diet is considered and should be the objective of future research.

As mentioned above, the finely minced meat products and canned meat contributed the most to total daily nitrite intake from processed meat products because of both the highest content of nitrites and the highest consumption among all analysed meat products categories (Table 4). These results are in accordance with the EFSA [19], where the most important contributors to the total cumulative exposure in all age groups were sausages and preserved meat, while pastes, pates, and terrines and meat specialities contributed less. In a recent US study [41], cured, cooked sausages, and whole-muscle brine-cured products were also the most important contributors to processed meat nitrite intake among children. In addition, in a recent Italian study, cooked ham and wurstel contributed the most to total EDI from processed meat products [19]. In the study from Korea, ham, sausage, and bacon were found to contribute the most to total daily nitrite intake from processed meat products [50].

Our data indicate need for changes in the consumption habits and legislation to decrease the nitrite exposure from processed meat products, particularly in the population of younger children. For example, the recent Estonian study showed that the exposure of children to nitrites declined over last 10 years as a result of changes in food preferences and decreased usage of nitrite in cured meat products by meat industries [48]. In line with that, the Denmark government in 2010 reduced maximal permitted levels of nitrites in processed meat food products to 60 mg/kg, leading to a much lower exposure (almost halved) than in other EU countries [19].

High exposure to nitrites in the long-term can be connected with several health risks. Particularly young children population may be on significant risk for long term over-consumption. Some harmful effects of nitrites on humans, such as methemoglobinemia (leading to cyanosis or anaemic hypoxia, which can be potentially lethal, especially in infants and children, who have lower nicotinamide adenine dinucleotide (NADH) cyb5r reductase activity, which converts methaemoglobin to haemoglobin), and cancerogenic nitrosamines formation, are well documented [19,51]. Related to cancerogenic nitrosamines, in 2010 the IARC, declared, “There is sufficient evidence in experimental animals for the carcinogenicity of nitrite in combination with amines or amide. There is limited evidence in experimental animals for the carcinogenicity of nitrite per se. Ingested nitrate and nitrite under conditions that result in endogenous nitrosation is probably carcinogenic to humans (Group 2A)” [52], while in 2018, the IARC re-evaluated their statements and declared: “There is sufficient evidence in humans for the carcinogenicity of consumption of processed meat. Consumption of processed meat causes cancer of the colorectum. Positive associations have been observed between consumption of processed meat and cancer of the stomach. There is inadequate evidence in experimental animals for the carcinogenicity of consumption of processed meat. Consumption of processed meat is carcinogenic to humans (Group 1)” [53]. Even though there is sufficient evidence for colorectal carcinoma, there is still insufficient evidence for other cancers: oesophageal, gastric, lung, non-Hodgkin’s lymphoma, thyroid, pancreatic, liver, ovarian, prostate, bladder, renal cancer, and brain tumours and glioma [19]. Furthermore, there is no sufficient evidence for risk of diabetes type 1 [54]. Nevertheless, some clinical studies have also demonstrated beneficial effects of dietary nitrates and nitrites, through their conversion to nitric oxide, especially in cardiovascular, immune, metabolic, and neural and reproductive health, and consumption of nitrites above the legislative limits are questioned as being less harmful [51,55].

### 3.4. Phosphorus Content in Processed Meat Products

Phosphorus content in different types of meat products is shown in Table 6. The results obtained are presented as mean, quartiles (Q), median values (Q2), the 95th percentile, and range. As seen in the table, the highest average and highest phosphorus content were observed in smoked meat products (6.12 ± 1.33 g/kg, 10.64 g/kg, respectively), while the lowest average values were obtained for liver sausage and pate (2.71 ± 1.05 g/kg). The average content of phosphorus (5.18 ± 1.31 g/kg) varied within a range between 0.27 and 10.64 g/kg. Analysis of the frequency distribution of phosphates contents (Figure 4) shows a normal distribution, and consequently, 95% of the data are between the average (SD ± 1.31) standard deviations. Of the 1900 retail samples we analysed, only 32 (1.7%) exceeded maximum limit for total phosphates expressed as P_2_O_5_ in meat products (<8 g/kg) [22]. In this study, the mean phosphate content in all groups of meat products was below the legal limit (<8 g/kg) [22]. Our results are in line with data obtained in previous studies [56,57]. 

Due to their a multifunctioning property, the addition of phosphates, as well as their blend, is a standard practice in the meat industry. Phosphates increase the water-holding capacity and, consequently, reduce drip loss and cooking loss [58]. The most common categories of products that use phosphates are cooked sausages, hams, and other whole-muscle products. This may be attributed to the fact that moisture retention is an important parameter of their quality. The addition of phosphates increases water holding in cooked sausages by 30–40 g per 100 g of meat [59]. On the other hand, phosphates are sometimes used in products, such as cured meat, to reduce cured colour development. In other products such as bacon, phosphates are used to improve texture during cooking by the consumer [60]. Yet, the use of phosphates at higher levels can produce a rubbery texture or impair the sensory characteristics of products such as giving the meat a soapy flavour [61]. In many countries, including Serbia, meat processing plants do not have an obligation to indicate the amount of phosphorus contained in (or added to) their products. However, comparing results observed in the present study with the information available in the literature, it could be concluded that the use of food-grade phosphates as a food additive in Serbia is generally in line with existing recommendations (up to 0.5%). Despite the fact that phosphates are widely used to improve quality characteristics of meat products, due to potential risks associated with chronic kidney disease (CKD), bone disease, and cardiovascular disease (CVD), phosphate reduction is a growing issue for meat producers.

### 3.5. Contribution of Processed Meat Products to the Daily Intake of Phosphorous 

Daily intake of phosphorus through processed meat products in child populations is expressed as EDI: at the mean, 2.06 mg/kg bw/day, and at P95th level l–2.73 mg/kg bw/day, contributed 5.14% and 6.82% to ADI, respectively (Table 7). There was a slightly lower phosphorus intake per kilogram body weigh in the children aged 1–3 years, compared with the children aged 3–9 years, with no observable difference between males and females. Major sources of phosphorus in meat products in the observed study sample were finely minced cooked sausages, canned meat products, and bacon (Figure 5), which were, at the same time, the most consumed processed meat products by children. 

Even though these EDI values are far below ADI, it should be considered that other studies on the dietary assessment of phosphorus report on excessive intake of phosphorus from the whole diet [62,63], and particularly from meat and meat products [64], and that phosphorus as additive accounts only as a part of total phosphorus content in these food groups [65]. The EFSA Panel from 2019, which analysed total phosphorus exposure in children, concluded that when the whole diet is regarded, the intake of phosphorus would exceed the ADI of 40 mg/kg bw/day in children aged 1–9 years, both at the mean and high exposure levels [28]. The report represented the results from 15 EU countries (including Austria, Belgium, Bulgaria, Czech Republic, Denmark, Finland, France, Germany, Greece, Italy, Latvia, Netherlands, Spain, Sweden, and UK), involving 13637 children (the same as for the nitrite Panel). The mean phosphorus EDI from all dietary sources was 69.70 (55–74) mg/kg bw/day in the 1–3-year-old group and 49.92 (33–62) mg/kg bw/day in the 3–9 age group. The highest exposure for the 1–3-year-old group was noted in Denmark and UK and in Finland for the 3–9-year-old group. 

The main contributing food categories to total phosphate EDI were bread, rolls and fine bakery wares, processed cheese, meat products, and sugars and syrups. Processed meat products contributed to the whole diet phosphorus exposure, at 6.1–14.0% in children 1–3 years old and 5.2–17.8% in children 3–9 years old [28]. According to these data, we could also predict that in Serbian children, the ADI could be exceeded if the all dietary sources of phosphates are considered. Thus, future studies should focus on determination of total phosphorus intake from the whole diet in this vulnerable population in Serbia. 

The EFSA Panel from 2019 concluded that phosphates have “low acute oral toxicity and there is no concern with respect to genotoxicity and carcinogenicity”, and that they “do not present any risk for reproductive or developmental toxicity” [28]. However, even though phosphorus is an essential element in the human body, excessive consumption of phosphorus can be related to several health risks, including cardiovascular complications—impaired endothelial function and hypertension, vascular calcifications, left ventricular hypertrophy, heart failure, and atrial fibrillation, as well as increased cardiovascular mortality [28]. High phosphorus intake can disrupt the hormonal regulation of phosphorus, calcium, and vitamin D and stimulate excessive secretion of parathormone and fibroblast growth factor-23 (FGF-23), leading to impaired bone turnover and bone demineralization, osteoporosis, bone fractures, and, due to higher renal tubular exposure to phosphates, tubular nephropathy (disseminated atrophy and tissue necrosis) and nefrocalcinosis, leading to renal impairment. Fibroblast growth factor-23 is directly related to cardiovascular events and has been associated with left ventricular hypertrophy, atrial fibrillation, heart failure, and cardiovascular mortality. Some studies have shown that FGF23 can directly induce left ventricular hypertrophy. However, there are insufficient data in humans to confirm all these health risks, and more research is needed [28]. Nevertheless, in cases of chronic renal disease with decreased renal function or vitamin D intoxication and low calcium dietary intake, gut absorption of phosphorus can increase. Phosphorus from food additives has the highest bioavailability because the inorganic sodium and potassium phosphate salts dissociate easily and do not require release by luminal phosphatases, making them thus more easily absorbed than phosphorus from organic phosphate in natural, unprepared animal or plant foods. Phosphorus from plants sources is less digestible and less bioavailable than phosphorus from animal sources because of complexes with phytates; bioavailability of phosphate from phytates is only 20–30%, since humans lack the enzyme phytase. Different cooking methods or industrial food processing can affect bioavailability and absorption, as well as the consumption of vitamin D and phosphate binders (calcium, magnesium, some amino acids), which can increase or decrease phosphate absorption. 

Considering all these potential health risks of excessive phosphates intakes, efforts should be made to decrease exposure to phosphates through additives. The meat industry should consider other technological approaches to preservation, such as application of ultrasound (US), high-pressure processing (HPP), and pulsed electric field (PEF), that modify the protein structure and improve its functional properties, allowing for the reduction of the content of additives in meat products [66].

### 3.6. Study Strengths and Limitations

This study on nitrites and phosphorus exposure risk assessment through meat products was conducted on a representative sample of the Serbian child population using harmonized food consumption data collected according to EFSA guidance methodology (within the EU Menu project survey), which makes these data comparable with other harmonized food consumption data in the whole of Europe. These data allow longitudinal monitoring and exposure assessment.

One of the study limitations is that only 2 days 24 h food records were explored, while it would be better to have at least 3 days, ideally 7 days, to cover interday variation in the intakes [67]. However, we performed FPQ data analysis on frequency of consumption, and the results were congruent with the 24 h food records data, indicating that frequency of consumption was well presented by the twice repeated 24 h records in studied population. Additionally, the study sample was stratified to proportionally cover weekdays and weekend days, capturing variability of dietary patterns. According to EFSA EU Menu methodology, food consumption data collection studies are performed using twice repeated 24 h records [33]. Total exposure from nitrites and phosphate—total EDI—could not be extracted in this study, as only consumption of processed meat products was observed. Further research should include other food groups, i.e., perform exposure risk assessment on the total diet.

## 4. Conclusions

The present study has shown that the content of nitrites in meat products in Serbia is within MPL, with only one product exceeding the limit. Intake of nitrites in the child population, on average, is below ADI. However, when intake was accounted for those who consumed meat products, these values indicated that intake of nitrites only from processed meat products was alarmingly high and exceeded ADI in a substantial number of consumers, particularly among the youngest children.

The content of phosphorus in analysed meat products was within MPL, with 3.7% of all processed meat products exceeding this value. Assessed intake of phosphorus from processed meat products was far below ADI values, even when regarded for the highest exposure scenario, which does not raise a concern. However, this finding should be taken with caution, as phosphorus from processed meat products is just one part of the total phosphorus intake from various dietary sources.

Taking into account the cumulative effect of long-term consumption, these findings raise a concern of exposure risk of nitrites in the child population. The results highlighted that intake of processed meat is high and frequent among this population group, which calls for attention in the food industry to consider reducing content of these additives in meat products. As learned from other countries, such as Denmark and Korea, lowering of MPL for nitrites resulted in gradual reduction of nitrite intake in a population. Decreased usage of nitrite in processed meat products by meat industries is recommended, especially for those products which are majorly consumed in child populations—finely minced cooked sausages and canned meat. 

Regulatory authorities should be informed that there is a potential risk and have their attention called to lower MPL for nitrites. The system for continual dietary exposure monitoring for these and other additives should be established in the country. On a wider perspective, these findings suggest that the food industry should move towards applying healthier, “greener” technologies for preservation, as the actual transformation of food systems towards nutrition-sensitive and sustainability require. Finally, rising awareness on harmful effect of nitrites in processed meat can bring a major shift in dietary preferences in children and their parents and caregivers.

## Figures and Tables

**Figure 1 nutrients-14-00242-f001:**
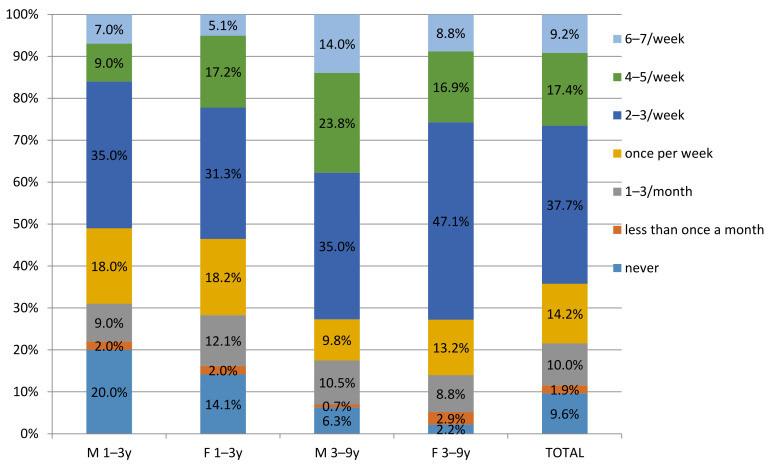
Frequency of consumption of meat products in children 1–9 in Serbia by Food Propensity Questionnaire (FPQ).

**Figure 2 nutrients-14-00242-f002:**
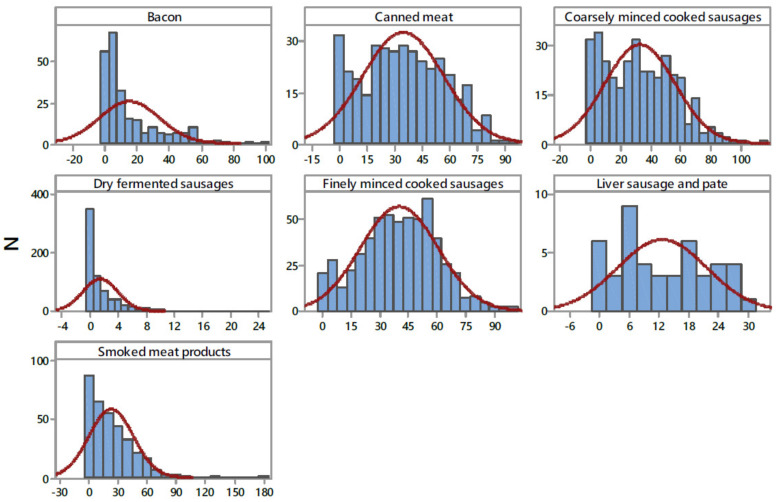
Distribution of NaNO_2_ content in examined processed meat products (mg kg^−1^). N—number of samples.

**Figure 3 nutrients-14-00242-f003:**
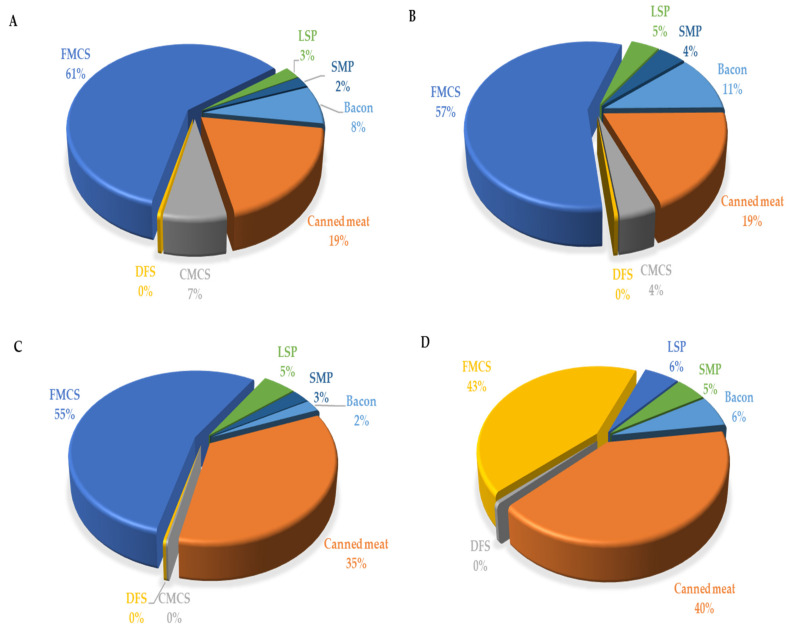
Contribution (%) of the most important food groups of processed meat products to dietary exposure of toddlers 1–3 years ((**A**)—male and (**B**)—female) and children 3–9 years ((**C**)—male and (**D**)—female), to nitrite ion (NO_2_^−^). CMCS—coarsely minced cooked sausages, DFS—dry fermented sausages, FMCS—finely minced cooked sausages, LSP—liver sausage and pate, SMP—smoked meat products.

**Figure 4 nutrients-14-00242-f004:**
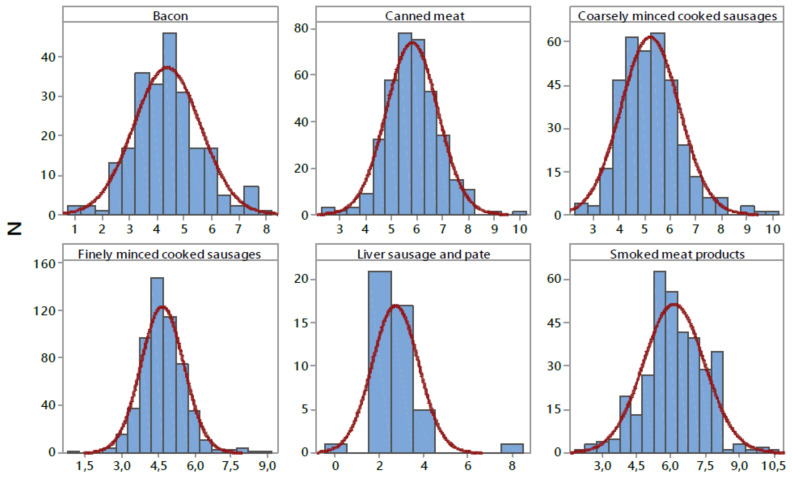
Distribution of phosphorus content (P_2_O_5_) in examined processed meat products (g kg^−1^). N—number of samples.

**Figure 5 nutrients-14-00242-f005:**
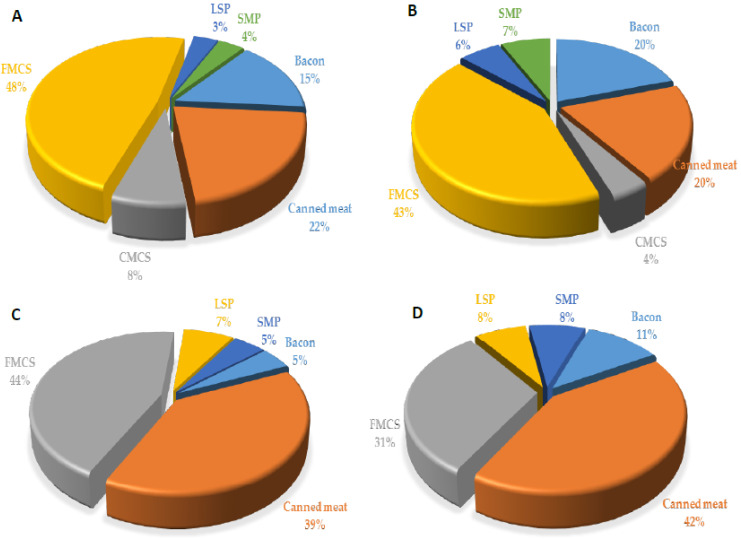
Contribution (%) of the most important food groups of processed meat products to dietary exposure of toddlers 1–3 years ((**A**)—male and (**B**)—female) and children 3–9 years ((**C**)—male and (**D**)—female) to phosphorus ion. CMCS—coarsely minced cooked sausages, FMCS—finely minced cooked sausages, LSP—liver sausage and pate, SMP—smoked meat products.

**Table 1 nutrients-14-00242-t001:** Characteristics of the study sample.

Age Group	*N*		Age (Year) Median (ICR)		Body Weight (kg) Mean ± SD	
	Male	Female	Male	Female	Male	Female
Toddlers, 1–3 years	149	141	2.0 (1.5–2.6)	2.0 (1.4–2.5)	13.92 ± 2.81	13.92 ± 2.76
Children, 3–9 years	147	139	6.1 (4.4–7.9)	6.2 (5.0–7.8)	23.99 ± 6.98	23.89 ± 6.63
Total sample	576					

*N*—number of participants; ICR—interquartile range

**Table 2 nutrients-14-00242-t002:** The average intake of processed meat products by children population (g/day).

Age Group		Bacon	Canned Meat	CMCS	DFS	FMCS	LSP	SMP	Total
		Mean ± SD (g/day)							
Toddlers, 1–3 years	M	2.17 ± 5.55	2.37 ± 6.40	0.92 ± 6.79	1.06 ± 3.90	6.49 ± 16.41	0.86 ± 3.42	0.40 ± 2.38	14.28 ± 44.86
	F	2.87 ± 12.71	2.17 ± 6.08	0.50 ± 3.08	1.13 ± 4.25	5.68 ± 14.41	1.41 ± 4.20	0.75 ± 3.03	14.51 ± 47.76
Children, 3–9 years	M	1.24 ± 5.37	8.01 ± 13.74	0.05 ± 0.64	1.75 ± 6.18	11.11 ± 19.71	3.22 ± 7.51	0.95 ± 3.89	26.33 ± 57.06
	F	2.93 ± 10.00	8.32 ± 11.51	--	2.65 ± 11.53	7.66 ± 18.74	3.21 ± 7.78	1.54 ± 7.34	26.30 ± 66.89
Average		2.29 ± 8.89	5.20 ± 10.39	0.37 ± 3.80	1.64 ± 7.10	7.76 ± 17.53	2.16 ± 6.11	0.90 ± 4.54	20.31 ± 58.35

M—male, F—female, CMCS—coarsely minced cooked sausages, DFS—dry fermented sausages, FMCS—finely minced cooked sausages, LSP—liver sausage and pate, SMP—smoked meat products.

**Table 3 nutrients-14-00242-t003:** Mean levels and ranges of nitrite content expressed as NaNO_2_ in examined processed meat products (mg/kg).

Meat Product	*N*	*n* (%)	Mean ± SD (mg/kg)	Q1 (mg/kg)	Q2 (mg/kg)	Q3 (mg/kg)	P95 (mg/kg)	Range (mg/kg)	MPL (mg/kg) [23]
Bacon	241	228 (94.6)	15.26 ± 18.74	2.74	6.78	21.00	53.66	0.33–100.38	
Canned meat	362	348 (96)	34.95 ± 22.12	18.00	33.77	52.28	71.1	0.08–93.45	
Coarsely minced cooked sausages	353	345 (97.7)	32.85 ± 23.25	11.31	31.31	51.01	71.14	0.05–113.51	
Dry fermented sausages	683	374 (54.7)	1.50 ± 2.48	0.00	0.41	1.95	6.15	0.05–24.88	150
Finely minced cooked sausages	580	578 (99.6)	40.25 ± 20.37	26.75	40.72	54.67	71.75	0.09–101.19	
Liver sausage and pate	46	42 (91.3)	12.62 ± 9.01	5.10	11.07	19.95	26.12	0.84–30.39	
Smoked meat products	338	322 (95.2)	23.09 ± 22.80	4.67	18.51	35.10	61.70	0.10–180.25	
Average	2603	2238 (85.97)	23.31 ± 23.75	1.72	15.97	41.08	67.10	0.05–180.25	

*N*—total number of analysed samples; *n*—number of samples that contained nitrite (%); 1st quartile (Q1), 25% of the data are less than or equal to this value; 2nd quartile (Q2), the median 50% of the data are less than or equal to this value; 3rd quartile (Q3), 75% of the data are less than or equal to this value; P95-95th percentile. MPL—maximum amount of nitrites that may be added during manufacturing [23]; LOQ—limit of quantification = 0.03 mg/kg.

**Table 4 nutrients-14-00242-t004:** EDI and risk characterization of nitrite ion (NO_2_^−^) intake through processed meat products.

Meat Product	Age Group	Gender	ADC (g/day)	Nitrite Ion (NO_2_^−^) Content		EDI (mg/kg bw/day)		Contribution to ADI (%)		ADI (mg/kg bw/day)
				Mean ± SD(mg/kg)	P95th (mg/kg)	Mean	P95th	Mean	P95th	
Bacon	Toddlers, 1–3 years	M	2.17	10.17 ± 12.50	35.77	0.002	0.006	2.26	7.97	0.07
		F	2.87			0.002	0.007	3.00	10.55	
	Children, 3–9 years	M	1.24			0.001	0.002	0.75	2.64	
		F	2.93			0.001	0.004	1.78	6.27	
Canned meat	Toddlers, 1–3 years	M	2.37	23.30 ± 14.75	47.4	0.004	0.008	5.67	11.54	
		F	2.17			0.004	0.007	5.18	10.53	
	Children, 3–9 years	M	8.01			0.008	0.016	11.12	22.62	
		F	8.32			0.008	0.017	11.59	23.58	
Coarsely minced cooked sausages	Toddlers, 1–3 years	M	0.92	21.90 ± 15.50	47.43	0.001	0.003	2.07	4.49	
		F	0.50			0.001	0.002	1.12	2.42	
	Children, 3–9 years	M	0.05			0.000	0.000	0.07	0.15	
		F	0.00			0.000	0.000	0.00	0.00	
Dry fermented sausages	Toddlers, 1–3 years	M	1.06	1.00 ± 1.65	4.11	0.000	0.000	0.11	0.45	
		F	1.13			0.000	0.000	0.12	0.48	
	Children, 3–9 years	M	1.75			0.000	0.000	0.10	0.43	
		F	2.65			0.000	0.000	0.16	0.65	
Finely minced cooked sausages	Toddlers, 1–3 years	M	6.49	26.83 ± 13.58	47.82	0.013	0.022	17.88	31.87	
		F	5.68			0.011	0.020	15.65	27.90	
	Children, 3–9 years	M	11.11			0.012	0.022	17.75	31.64	
		F	7.66			0.009	0.015	12.29	21.90	
Liver sausage and pate	Toddlers, 1–3 years	M	0.86	8.41 ± 6.0	17.41	0.001	0.001	0.74	1.54	
		F	1.41			0.001	0.002	1.22	2.52	
	Children, 3–9 years	M	3.22			0.001	0.002	1.61	3.34	
		F	3.21			0.001	0.002	1.61	3.34	
Smoked meat products	Toddlers, 1–3 years	M	0.40	15.39 ± 15.19	41.13	0.000	0.001	0.62	1.67	
		F	0.75			0.001	0.002	1.18	3.15	
	Children, 3–9 years	M	0.95			0.001	0.002	0.87	2.32	
		F	1.54			0.001	0.003	1.42	3.78	
Average	Toddlers, 1–3 years	M	2.04	15.54 ± 15.84	44.86	0.003	0.006	4.20	8.50	
		F	2.07			0.003	0.006	3.92	8.22	
	Children, 3–9 years	M	3.76			0.003	0.006	4.61	9.02	
		F	3.76			0.003	0.006	4.12	8.50	
Total	Toddlers, 1–3 years	M	14.28	15.54 ± 15.84	44.86	0.021	0.042	29.37	59.52	
		F	14.51			0.019	0.040	27.46	57.54	
	Children, 3–9 years	M	26.33			0.023	0.044	32.27	63.12	
		F	26.30			0.020	0.042	28.85	59.52	

Nitrate is expressed as nitrate ion (66.65% of NaNO_2_); ADC–average daily consumption of meat products (Table 1); EDI–estimated daily intake; ADI–acceptable daily intake [19,24].

**Table 5 nutrients-14-00242-t005:** EDI and risk characterization of nitrite ion (NO_2_^−^) intake in consumers of processed meat products.

Age Group	Gender	*N*	% of Total Subjects	EDI (mg/kg bw/day)		Contribution to ADI (%)		N (%) Above ADI	
				Mean	P95th	Mean	P95th	Mean	P95th
Toddlers, 1–3 years	M	81	54.36	0.038	0.076	53.89	109.00	10 (12.35)	32 (39.51)
	F	81	57.45	0.037	0.074	52.37	106.18	10 (12.35)	31 (38.27)
Children, 3–9 years	M	113	76.87	0.021	0.044	30.33	62.57	8 (7.08)	17 (15.04)
	F	111	79.86	0.021	0.044	30.64	63.29	8 (7.21)	18 (16.22)
Average/Total		386	67.01	0.028	0.057	39.99	81.67	36 (9.33)	98 (25.39)

**Table 6 nutrients-14-00242-t006:** Mean levels and ranges for phosphorus (P_2_O_5_) in processed meat samples (g/kg).

Meat Product	*N*	Mean ± SD (g/kg)	Q1 (g/kg)	Q2 (g/kg)	Q3 (g/kg)	P95 (g/kg)	Range (g/kg)	Above MPL (%)
Bacon	230	4.41 ± 1.22	3.57	4.39	5.13	6.53	1.10–7.94	--
Canned meat	375	5.79 ± 1.01	5.14	5.76	6.43	7.40	2.37–9.83	6 (1.6)
Coarsely minced cooked sausages	353	5.21 ± 1.14	4.44	5.12	5.89	7.14	2.25–9.92	7 (2)
Finely minced cooked sausages	551	4.70 ± 0.88	4.16	4.64	5.17	6.10	1.12–9.22	4 (0.7)
Liver sausage and pate	45	2.71 ± 1.05	2.18	2.50	3.08	3.88	0.27–7.96	--
Smoked meat products	346	6.12 ± 1.33	5.35	6.02	6.97	7.98	2.11–10.64	15 (4.3)
Total	1900	5.18 ± 1.31	4.36	5.11	5.98	7.54	0.27–10.64	32 (1.7)

MPL—maximum permitted level (<8 g/kg) [22].

**Table 7 nutrients-14-00242-t007:** EDI of phosphorus and risk characterization of phosphorus intake (mg/kg bw/day).

Meat Product	Age Group	Gender	ADC (g/day)	Phosphorus Content		EDI (mg/kg bw/day)		Contribution to ADI (%)		ADI (mg/kg bw/day)
				Mean ± SD (g/kg)	P95th (g/kg)	Mean	P95th	Mean	P95th	
Bacon	Toddlers, 1–3 years	M	2.17	1.92 ± 0.53	2.85	0.30	0.44	0.75	1.11	40
		F	2.87			0.40	0.59	0.99	1.47	
	Children, 3–9 years	M	1.24			0.10	0.15	0.25	0.37	
		F	2.93			0.24	0.35	0.59	0.87	
Canned meat	Toddlers, 1–3 years	M	2.37	2.53 ± 0.44	3.23	0.43	0.55	1.08	1.38	
		F	2.17			0.39	0.50	0.98	1.26	
	Children, 3–9 years	M	8.01			0.84	1.08	2.11	2.70	
		F	8.32			0.88	1.12	2.20	2.81	
Coarsely minced cooked sausages	Toddlers, 1–3 years	M	0.92	2.27 ± 0.49	3.11	0.15	0.21	0.38	0.52	
		F	0.50			0.08	0.11	0.20	0.28	
	Children, 3–9 years	M	0.05			0.00	0.01	0.01	0.02	
		F	0.00			0.00	0.00	0.00	0.00	
Finely minced cooked sausages	Toddlers, 1–3 years	M	6.49	2.05 ± 0.38	2.66	0.96	1.24	2.39	3.10	
		F	5.68			0.84	1.09	2.09	2.72	
	Children, 3–9 years	M	11.11			0.95	1.23	2.37	3.08	
		F	7.66			0.66	0.85	1.64	2.13	
Liver sausage and pate	Toddlers, 1–3 years	M	0.86	1.18 ± 0.46	1.7	0.07	0.11	0.18	0.26	
		F	1.41			0.12	0.17	0.30	0.43	
	Children, 3–9 years	M	3.22			0.16	0.23	0.40	0.57	
		F	3.21			0.16	0.23	0.40	0.57	
Smoked meat products	Toddlers, 1–3 years	M	0.40	2.67 ± 0.58	3.48	0.08	0.10	0.19	0.25	
		F	0.75			0.14	0.19	0.36	0.47	
	Children, 3–9 years	M	0.95			0.11	0.14	0.26	0.34	
		F	1.54			0.17	0.22	0.43	0.56	
Average	Toddlers, 1–3 years	M	2.04	2.26 ± 0.57	3.30	0.28	0.38	0.71	0.95	
		F	2.07			0.28	0.38	0.70	0.95	
	Children, 3–9 years	M	3.76			0.31	0.40	0.77	1.01	
		F	3.76			0.30	0.40	0.75	0.99	
Total	Toddlers, 1–3 years	M	14.28	2.26 ± 0.57	3.30	1.99	2.65	4.97	6.62	
		F	14.51			1.97	2.65	4.93	6.62	
	Children, 3–9 years	M	26.33			2.16	2.83	5.40	7.07	
		F	26.30			2.10	2.78	5.26	6.95	

ADC—average daily consumption of meat products; ADI—acceptable daily intake [28].

## Data Availability

Not applicable.

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
