# Peer review of "The Intake of Phosphorus and Nitrites through Meat Products: A Health Risk Assessment of Children Aged 1 to 9 Years Old in Serbia"

_nutrients, 2022, doi:10.3390/nu14020242_

Round 1
Reviewer 1 Report
This study provides interesting findings on nitrites and phosphorus exposure throug meat products in young children. Commercially availabe meat products were analyzed for nitrate and phosphorous content. Although the EDI of Nitrites from meat products exceeded the ADI, the EDI of phosphorus from meat was below the ADI. Further studies should focus on the exposure and risk assessment on total diet. The paper is well written, below some minor comments.
-The abbreviation for colorectal cancer is CRC however in the abstract both CAC and CRC are used.
-Table 1 shows the mean age of the children, the standard deviation is large, wouldn't it be better to use the median and IQR? (it seems that age is not normally distributed)
Author Response
-The abbreviation for colorectal cancer is CRC however in the abstract both CAC and CRC are used.
Response: instead CAC abbreviation, CRC was used
-Table 1 shows the mean age of the children, the standard deviation is large, wouldn't it be better to use the median and IQR? (it seems that age is not normally distributed).
Response: We amended the table, and now the age is expressed as median and IQR as requested.

Reviewer 2 Report
The aim of the paper is to estimate dietary exposure of the Serbian children population to nitrites and phosphorus through meat product. The research problem undertaken in the work is interesting, although the work requires correction before publication. Overall, the title and abstract reflects the manuscript content. Keywords are well-chosen, as well. However, I indicate several suggestions to be considered by the authors to improve the manuscript.
First of all, there is no full justification why only meat products are included as a source of nitrite exposure, since according to available data, meat products are not the main source of nitrite in human diet in many European countries (e.g. https://www.mdpi.com/2076-3921/9/3/241). Additionally, why was nitrates not included, the more so because the authors of the manuscript frequently refer to both nitrates and nitrites.
Line 138 - the authors write that nitrates and nitrites have been analyzed!!!
Lines 295 - nitrate or nitrite
The reference to the legislative data requires clarification (Line 82-85). The context of the sentence may indicate that the stated limits for nitrite in the EU refer to the residues in the product while they refer to the level of addition. Similarly, in Table 3, the MPL applies to the limits in the EU or in Serbia?
In introduction, where the authors refer to the high consumption of meat products by consumers in Serbia (lines 118-120), it is worth providing statistical data presenting the average annual consumption.
Lines 60-61 - The authors refer to many studies and give only one reference. Should be completed with others.
The entire text requires proofreading by a native speaker, sentences are in many places with linguistic errors (e.g. line 58). In addition, there are punctuation errors in many places, e.g. line 64.
Author Response
Ad. 1. First of all, there is no full justification why only meat products are included as a source of nitrite exposure, since according to available data, meat products are not the main source of nitrite in human diet in many European countries (e.g. https://www.mdpi.com/2076-3921/9/3/241). Additionally, why was nitrates not included, the more so because the authors of the manuscript frequently refer to both nitrates and nitrites.
Response: The answer to the both query, lies in the technological, regulatory and food safety aspects. Also, other factors could be explained.
- This is not total diet study. Despite meat products are not the main source of nitrite in human diet in many European countries, monitoring of food additives is mandatory in EU, but not in the Serbia. For this purpose/exposure assessment, meat products were taken into the consideration as a contributor of nitrites and phosphorus intake. This is the first study in the Serbia where are estimated dietary exposure of the Serbian population to nitrites and phosphorus through meat product.
Although the vast majority of consumed nitrates and nitrites come from natural vegetables and fruits rather than food additives, there is currently a great deal of consumer pressure for the production of meat products free of or with reduced quantities of these compounds (https://www.mdpi.com/2076-3921/9/3/241).
- From technological point of view, only nitrite exerts a positive effect on the meat products, making them safe for consumption and extending their shelf-life. Therefore, nitrite salt has been more frequently used in meat industry than nitrates.
- From regulatory point of view, content of added nitrate is reasonable to analyze only in fresh products before ripening or processing, while residual amount of nitrite, at the end of the production process should be analyzed. Considering that, during production phases, nitrates are reduced to nitrites, a small amount of nitrate could be expected, while residual nitrite can be detected. Nitrate content decrease during ripening as a consequence of degradation of the nitrates present by transformation into nitrites.
- Also, should be mentioned that Institute of meat hygiene and technology is officially authorized laboratory for analysis, super analysis and quality control mainly meat and meat products, collected from meat industry during self-control or official control.
Ad. 2. Line 138 - the authors write that nitrates and nitrites have been analyzed!!!
Response: mistake was corrected. word nitrate was omitted, due to only nitrite has been analysed.
Ad. 3. Lines 295 - nitrate or nitrite
Response: In section Results and Discussion, subsection Nitrate content in processed meat products, word nitrate was replaced with nitite, where are necessary.
Ad. 4. The reference to the legislative data requires clarification (Line 82-85). The context of the sentence may indicate that the stated limits for nitrite in the EU refer to the residues in the product while they refer to the level of addition. Similarly, in Table 3, the MPL applies to the limits in the EU or in Serbia?
Response: The statement of reviewer is correct. According to EU legislation the amount of nitrite was related to added nitrite. A similar regulation has entered into force and is currently applicable in Serbia. Therefore, statements were replaced accordingly. Although, reference related to the (MPL) maximum amount of nitrites that may be added during manufacturing was inserted below of the table, ref. number was also inserted in the table [23]. Probably is now clearer.
Ad. 5. In introduction, where the authors refer to the high consumption of meat products by consumers in Serbia (lines 118-120), it is worth providing statistical data presenting the average annual consumption.
Response: Please read carefully, sentence where it is written „Meat products such as ham, sausages, bacon, frankfurters, salami, etc are widely consumed by all groups of the population in Serbia, at home, or in fast foods restaurant”.
Ad. 6. Lines 60-61 - The authors refer to many studies and give only one reference. Should be completed with others.
Response: Considering that most comprehensive studies has been conducted by World Cancer Research Fund and IARC, sentence was re-writhen to include given references as main source of data.

Round 2
Reviewer 2 Report
The authors responded to most of the comments, although there are still typos, e.g. unnecessary dots before the footnotes (lines 65, 78)